# Most Women with Previous Gestational Diabetes Mellitus Have Impaired Glucose Metabolism after a Decade

**DOI:** 10.3390/ijms19123724

**Published:** 2018-11-23

**Authors:** Wahlberg Jeanette, Ekman Bertil, Arnqvist J. Hans

**Affiliations:** 1Department of Endocrinology and Department of Medical and Health Sciences, Linköping University, S-581 85 Linköping, Sweden; Jeanette.Wahlberg@regionostergotland.se; 2Department of Endocrinology and Department of Clinical and Experimental Medicine, Linköping University, S-581 85 Linköping, Sweden; hans.arnqvist@liu.se

**Keywords:** gestational diabetes mellitus, maternal obesity, insulin resistance, impaired fasting glucose, impaired glucose tolerance, OGTT, fasting glucose, proinsulin, C-peptide, autoimmunity

## Abstract

Of 1324 women diagnosed with gestational diabetes mellitus (GDM) in Sweden, 25% reported >10 years after the delivery that they had developed diabetes mellitus. We assessed the long-term risk of all glucose metabolic abnormalities in a subgroup of these women. Women (*n* = 51) previously diagnosed with GDM by capillary blood glucose ≥9.0 mmol/L (≈plasma glucose ≥10.0 mmol/L) after a 2 h 75 g oral glucose tolerance test (OGTT) were included. All underwent a clinical and biochemical evaluation, including a second 2 h 75 g OGTT. Individuals with known type 1 diabetes were excluded. At the follow-up, 12/51 (24%) reported previously diagnosed type 2 diabetes. Another four cases were diagnosed after the second OGTT, increasing the prevalence to 16/51 cases (31%). Impaired fasting plasma glucose (IFG) was diagnosed in 13/51 women and impaired glucose tolerance (IGT) in 10/51 women, leaving only 12 women (24%) with normal glucose tolerance. In addition, 2/51 women had high levels of glutamic acid decarboxylase (GAD) antibodies; of these, one woman classified as type 2 diabetes was reclassified as type 1 diabetes, and the second GAD-positive woman was diagnosed with IGT. Of the women diagnosed with GDM by a 2 h 75 g OGTT, a large proportion had impaired glucose metabolism a decade later, including type 1 and type 2 diabetes.

## 1. Introduction

Women with gestational diabetes (GDM) have a high risk of developing manifest diabetes mellitus [1,2,3,4,5], and a meta-analysis in 2009 reported a 7.5-fold increased risk of developing diabetes in women who were diagnosed with GDM during pregnancy [6]. The prevalence of GDM is generally proportional to the prevalence of underlying type 2 diabetes, and the risk for women with GDM to develop diabetes mellitus later in life depends on several factors, like follow-up time, insulin need during pregnancy, and ethnicity [7]. It is known that mothers with GDM are older and have a higher prevalence of obesity and chronic hypertensive disease than the background population [8,9]. GDM is also an independent risk factor for long-term cardiovascular morbidity [10]. As obesity and GDM are closely related, rising rates of overweightness and obesity worldwide may result in more women with GDM. However, very long-term follow-up studies, which have also been influenced by the obesity epidemic during recent years, are scarce [11]. We recently reported a prevalence of 25% of manifest diabetes mellitus in a large Swedish cohort of GDM women when followed up by a questionnaire 11 years after GDM diagnosis [12]. A weakness of questionnaire studies is that they do not provide biochemical data for the diagnosis of glucose intolerance, i.e., impaired fasting glucose (IFG), impaired glucose tolerance (IGT), or diabetes type validated with autoantibodies (glutamic acid decarboxylase, GAD). Despite the well-known risk of future diabetes in these women, follow-up has not been optimal: a newly published report from primary care units in England indicated that only 20% of GDM women had a regular follow-up [13]; a single-centre study recently reported that around 50% were followed up with oral glucose tolerance testing (OGTT) [14]; and in our study, 60% reported a follow-up after GDM diagnosis [12].

With the present study, we aimed to conduct a careful biochemical evaluation, including a 2 h 75 g OGTT, in a cohort of women in the southeast region of Sweden diagnosed with GDM 12 years earlier to assess the long-term risk of glucose metabolic abnormalities.

## 2. Results

### 2.1. Characteristics of Women with Previous GDM at Clinical Follow-Up (n = 51)

The mean age of the women in the study was 43 ± 6 years. Twelve women (24%) had BMI <25, 17 women (33%) had BMI in the range of 25–29.9, and 22 women (43%) had BMI ≥30. The numbers of pregnancies including the index pregnancy were one (*n* = 4, 8%), two (*n* = 25, 49%), three (*n* = 18, 35%), four (*n* = 3, 6%), and ten pregnancies (*n* = 1, 2%). Eight subjects (16%) smoked. During the GDM period, all subjects had diet therapy, and 14 individuals (27%) also had insulin treatment. Group data are given in Table 1 and Table 2 for subjects with normal glucose tolerance, manifest diabetes mellitus, IFG, and IGT.

### 2.2. Manifest Diabetes Mellitus

In all, 12/51 (24%) women reported diagnosed diabetes mellitus at the clinical follow-up. The median follow-up time for the women who reported manifest diabetes mellitus was 12.6 years and that for the group not reporting diabetes mellitus was 11.7 years. Eight women with diabetes mellitus were on diet treatment, while four subjects were treated with metformin. None were treated with insulin. Another 4/51 (8%) women were diagnosed with diabetes mellitus during the follow-up by means of 2 h 75 g OGTT, increasing the prevalence of diabetes mellitus to 16/51 subjects (31%) (Table 2).

### 2.3. Normal Glucose Tolerance, IFG, and IGT

Twelve subjects had normal glucose tolerance (24%), 13 (26%) were found to have IFG (6.1–6.9 mmol/L with a 2 h capillary plasma glucose value <8.9 mmol/L), and 10 (20%) had IGT (capillary plasma glucose 8.9–12.1 mmol/L after 2 h OGTT) at the clinical follow-up (Table 2). It is to be noted that one woman treated with bariatric surgery with normal fasting glucose of 4.7 had an inverse response to OGTT with a low 2 h plasma glucose value of 2.0 mmol/L, indicating a dumping syndrome. Another woman diagnosed with IGT reported previous pancreatitis, which could have been of importance for her impaired glucose tolerance as her BMI was only 21.

### 2.4. Heredity, Country of Origin, and Additional Pregnancies

The countries of origin of the women were Sweden (88%, *n* = 45), Europe outside Sweden (2%, *n* = 1), and outside Europe (10%, Middle East *n* = 4, and China *n* = 1). Overall heredity for diabetes mellitus was common: 30 (59%) women had a first-degree relative (parents or siblings) with diabetes mellitus and 12 (24%) had a second-degree relative with diabetes mellitus. In the group diagnosed with diabetes mellitus, all 16 subjects had a first-degree (81%) or second-degree relative with diabetes (19%). Table 1 shows the group data.

### 2.5. Other Diseases and Other Medication

Patients reported hypertension (*n* = 3), dyslipidemia (*n* = 3), and chronic pancreatitis (*n* = 1), and one patient had undergone a gastric bypass procedure. Six (12%) patients used contraceptive tablets, and 13 (25%) had a hormonal intrauterine device. Other medications used were lipid-lowering drugs (*n* = 3) and antihypertensive drugs (*n* = 2).

### 2.6. Characteristics of the Group with Manifest Diabetes Mellitus

Having a relative with diabetes mellitus was more common in the group with diabetes compared with in the group of subjects with normal glucose tolerance. There was no difference regarding social status (married/unmarried/divorced) or employment between the group with diabetes and the other groups. Waist circumference and sagittal abdominal diameter did not differ significantly between those with diabetes and those with normal glucose tolerance, but systolic (128 compared to 116 mmHg, *p =* 0.031) and diastolic (81 compared to 73 mmHg, *p =* 0.053) blood pressure tended to be higher in the diabetes group. There were no significant differences between subjects with diabetes mellitus and subjects without diabetes mellitus regarding cholesterol, triglycerides, and Apolipoprotein B/Apolipoprotein A (ApoB/ApoA) quotient. The group with diabetes mellitus was not significantly more overweight or obese than the other groups not diagnosed with diabetes mellitus. Seven women with diabetes mellitus had increased their weight from the beginning of the pregnancy until the follow-up visit at our clinic. There were not significantly more women that had increased their weight in the diabetes group compared to in the groups not diagnosed with diabetes. The development of diabetes mellitus after GDM was not related to any treatment they had during the gestational diabetes period (diet or insulin).

### 2.7. GAD Antibodies, ln(Proinsulin), and C-Peptide

Two of the 51 women tested had high levels of GAD antibodies; of these, one woman classified as type 2 diabetes was reclassified as type 1 diabetes, and the second GAD-positive woman was diagnosed with IGT. No differences were found between the four groups in terms of ln(proinsulin) and C-peptide levels (Table 2). The ln(proinsulin)/C-peptide quotient tended to be higher in women with diabetes compared to in nondiabetic women (*p =* 0.075).

Including all 51 subjects in parametric correlation analyses, both ln(proinsulin) and C-peptide correlated positively with weight, BMI, waist circumference, P-triglycerides, and systolic and diastolic blood pressure. Additionally, ln(proinsulin) (but not C-peptide) correlated positively with fasting P-glucose and 2 h OGTT plasma glucose, but negatively with plasma HDL-Cholesterol (Table 3).

## 3. Discussion

This study shows a very high prevalence of glucose metabolic abnormalities manifesting in the long run after GDM diagnosis with a 2 h 75 g OGTT and using capillary blood glucose ≥9.0 mmol/L (≈plasma glucose ≥10.0 mmol/L) as a cut-off for diagnosis. At the follow-up after a median of 12 years, one-third of our GDM population had developed diabetes mellitus; this is in line with other studies showing a diabetes prevalence of 30–40% at follow-up of women with GDM 5–15 years after the index pregnancy [3,4,11]. In addition to women with manifest diabetes, many women exhibit IGT or IFG—46% in our study, leaving only 24% with normal glucose tolerance. In a follow-up of GDM five years after pregnancy, Ekelund et al. found IFG/IGT in 22% of the women [3], and Lauenborg et al. [11] reported 27% IFG/IGT after a median follow-up time of 9.5 years. Several studies thus suggest that most women with GDM will have diabetes or glucose intolerance at a long-term follow-up. Both in our study and in the study by Linné et al. [4], several undiagnosed cases were found, underlining the need for regular testing to detect diabetes. The results of the follow-up with 2 h 75 g OGTT thus show that GDM diagnosed using the Diabetic Pregnancy Study Group (DPSG) diagnostic criteria [15,16] is a strong predictor for future prediabetes/diabetes mellitus. There is no consensus regarding the use of fasting plasma glucose, OGTT, or HbA1c when following up on GDM. Glucose abnormalities defined by both IFG and IGT are risk predictors for diabetes, even if IGT defines a much larger target population for prevention [17]. It should be noted that in our study, HbA1c could not distinguish between normal subjects and subjects with IGT and/or IFG. However, in the group with diabetes, HbA1c was elevated; it should therefore be considered for follow-up and is today generally recommended for the diagnosis of diabetes [18]. Overall, the high frequency of glucose abnormalities found in this and previous studies indicates that women with GDM should be offered lifelong regular glucose measurements.

The original reason for defining and detecting GDM was to be able to identify those women at risk of diabetes mellitus later in life, a risk which depends on several factors [5]. In the last consensus statement by the International Association of the Diabetes and Pregnancy Study Groups IADPSG [19], endorsed by the American Diabetes Association (ADA) [20] and World Health Organization (WHO) [21], GDM is diagnosed if one or more of the following glucose values is met or exceeded: a fasting venous plasma glucose ≥5.1 mmol/L and/or a 1 h value ≥10.0 mmol/L and/or 2 h value ≥8.5 mmol/L post 75 g OGTT. This recommendation identifies GDM in 16–18% of all women if applied to the Hyperglycemia and Adverse Pregnancy Outcome Study (HAPO) data [19]. This much higher prevalence of GDM than with older criteria like those used in the present study has been criticized, as the HAPO study was observational in design and cannot provide information regarding the effectiveness of treatment of women with glucose concentrations that are lower than the former thresholds for GDM diagnosis [22,23,24,25]. The IADPSG criteria for the diagnosis of GDM derived from the HAPO study lack data about the future risk of diabetes [26,27]. It is important to note that our results could not be directly applicable to these lower cut-off values for GDM used in the HAPO study.

Overweightness and metabolic syndrome are more common in GDM women [28,29]. The metabolic burden was high in our patients, with 33% overweight subjects and 43% obese individuals at follow-up. This high prevalence of remaining overweight/obese, together with the strong correlations found in our study between C-peptide/proinsulin and cardiovascular risk markers like body composition, lipids, and blood pressure, indicates that the GDM diagnosis overlaps with the metabolic syndrome to a great extent [30] and is potentially linked to the insulin-like growth factor (IGF) system [31].

Type 2 diabetes could to some extent be prevented by lifestyle interventions, while type 1 diabetes, which was 10 times more common after GDM (5%) than in the Swedish general population (0.5%) [12], is not preventable. We measured GAD antibodies in the present study, and despite excluding subjects with known type 1 diabetes, we found additional individuals with signs of autoimmunity against islet cells. Despite excluding women with known type 1 diabetes from the study, the finding of further GAD-positive women strengthens the data that around 5–6% in a Scandinavian GDM population are GAD positive postpartum [32]. These results, showing a high prevalence of type 1 diabetes in GDM women, could to some extent explain the results from a newly published Finnish study, which also found a high risk for diabetes in non-obese women five years after GDM [33].

The major strength of this study was the careful biochemical and clinical analysis of the women with previous GDM. We measured plasma glucose with a capillary technique as both capillary blood and venous plasma samples can be used for the diagnosis of diabetes mellitus, but are not considered to be interchangeable [34,35]. Reliable ambulant methods for capillary glucose speed up the diagnostic procedure for diabetes mellitus, are economically favorable compared with venous plasma glucose methods, and avoid the instability of glucose in plasma after blood sampling [36]. How representative the cohort of 51 women is of all GDM women in Sweden may be questioned [12]. We previously reported that high BMI, 2 h OGTT blood glucose, and having been born outside Europe are risk factors for the progression of GDM to overt diabetes [12]. Comparing the baseline data of the cohort of 51 women who were examined in this study with those of the larger group, we found that slightly more women were of non-European heritage in the larger group and their 2 h OGTT glucose was significantly higher (Table 1). This suggests that if there is any bias, we have underestimated the prevalence of diabetes.

In conclusion, GDM diagnosed from a capillary blood glucose level of ≥9.0 mmol/L (≈plasma glucose ≥10.0 mmol/L) after a 75 g OGTT indicates the later development of impaired glucose metabolism in most of the women diagnosed with GDM, and our data indicates that all women with prior GDM should be offered lifelong regular glucose measurements. Moreover, type 1 diabetes should be regarded as a diagnostic alternative.

## 4. Materials and Methods

### 4.1. Participants

In a prospective nationwide registration of GDM in Sweden, 2085 pregnancies in 2025 women were reported from 1 January 1995 to 31 December 1999 [12]. For the diagnosis of GDM, a 75 g oral glucose load was used and a 2 h capillary blood glucose value ≥9.0 mmol/L (≈plasma glucose ≥10 mmol/L) was taken as a GDM-positive result according to the European Association for the Study of Diabetes (EASD) study group for gestational diabetes [15,37]. Capillary B-glucose used for diagnosis at the maternal healthcare clinics was analysed with HemoCue^®^ (HemoCue Ltd., Ängelholm, Sweden), a method with a high precision and accuracy and total CV less than 5% [38]. The GDM cohort was followed up 8.5–13.5 years after initial diagnosis with a questionnaire, which was answered by 1324 GDM women (65%) [12].

We identified women (*n* = 195) from the Swedish cohort living in our regional area of the southeast of Sweden, near Linköping University Hospital (1 million inhabitants). Of these, the first questionnaire was answered by 146 women (75%). In the present study, we invited women from the original cohort of 195 individuals living within 100 km of Linköping University Hospital for clinical examination. Individuals with known type 1 diabetes (*n* = 7) were not included. In all, 51 women accepted the invitation to participate in the clinical part of the study (Table 4 and Figure 1).

### 4.2. Measurements

The 51 women invited were examined by way of fasting plasma glucose and a 75 g OGTT. Additional laboratory measurements, including lipids, blood pressure, pulse rate, weight, length, waist circumference, and sagittal abdominal diameter, were collected. Duplicate samples of 5 μL capillary blood were collected in HemoCue Glucose cuvettes and analysed in a HemoCue Glucose 201 Analyser (HemoCue Ltd., Ängelholm, Sweden) [38], which converts blood glucose concentrations to equivalent plasma glucose concentrations by multiplying by an adjustment factor of 1.117. For the diagnosis of diabetes mellitus, a capillary fasting plasma glucose level ≥7.0 mmol/L and/or a 2 h capillary plasma glucose value ≥12.2 mmol/L after a 75 g oral glucose load was considered as diabetes mellitus, while fasting plasma glucose <7.0 mmol/L and a 2 h capillary plasma glucose value ≥8.9 and <12.1 mmol/L was defined as impaired glucose tolerance (IGT). Impaired fasting glucose (IFG) was defined as a fasting plasma glucose between 6.1 and 6.9 mmol/L with a 2 h capillary plasma glucose value <8.9 mmol/L [39]. HbA1c was analysed using a TOSOH G7 automated hemoglobin analyser (Tosoh Bioscience, Tokyo, Japan). Proinsulin was measured using Mercodia Proinsulin ELISA (Mercodia, Uppsala, Sweden) and C-peptide using Mercodia C-peptide ELISA (Mercodia, Uppsala, Sweden), and GAD antibodies were measured as described by Kordonouri et al. [40]. Blood lipids were analysed using the routine method in the department of clinical chemistry. Blood pressure was measured in the supine position after a 5 min rest and the mean of two measurements in the right arm was recorded. Hypertension was defined as blood pressure >140/90.

Overweightness was defined as 25 ≤ BMI < 30 kg/m^2^ and obesity as BMI ≥ 30 kg/m^2^. A waist circumference in a standing position of >80 cm was considered a moderate risk and >88 cm a high risk. Data regarding the development of diabetes, treatment for diabetes mellitus, treatment during pregnancy, other diseases, smoking, any concomitant medication, follow-up after pregnancy, contraceptives, later pregnancies, heredity for diabetes, the mother’s birth weight, married/unmarried, and occupation were collected at the visit.

### 4.3. Ethics

The ethics committe at Linköping University identification code M147-04 (approval date 10 November 2004), approved the study. The participants in the study were informed about the purpose of the study and gave their written informed consent.

### 4.4. Statistical Analysis

Continuous variables are presented as mean ± standard deviation (SD) or median with interquartile range (IQ), as appropriate. Number and percentage are reported for categorical variables, and differences between groups were evaluated using the Chi-square test. Comparisons between groups of continuous data were made using an unpaired Student’s *t*-test. Statistical analysis of differences between three or more groups was performed using ANOVA with Bonferroni post hoc testing, significant at the 0.05 level. For the estimation of linear associations, the Pearson correlation coefficient was calculated. Values for serum proinsulin were transformed to their corresponding natural logarithm (ln) to accomplish a normal distribution. A significance level of *p* < 0.05 (two-sided) was used. Statistics were calculated on a PC using the Statistical package for Social Science (SPSS Statistics 23, IBM, Stockholm, Sweden).

## Figures and Tables

**Figure 1 ijms-19-03724-f001:**
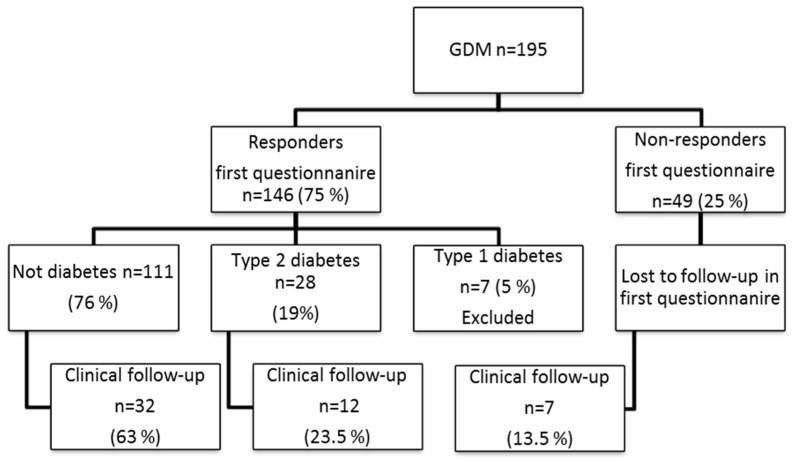
Clinical follow-up cohort, *N* = 51. Between 1995 and 1999, women with gestational diabetes mellitus (GDM, *n* = 2025) were reported in a Swedish nationwide study [12]. We identified women (*n* = 195) from the cohort living in our regional area of southeast Sweden, near Linköping University Hospital (1 million inhabitants). Of these, the first questionnaire was answered by 146 women (75%). In the present study, we invited women from the original cohort of 195 individuals living within 100 km of Linköping University Hospital for clinical examination. Individuals with known type 1 diabetes (*n* = 7) were not included. In all, 51 women accepted the invitation to participate in the clinical part of the study.

**Table 1 ijms-19-03724-t001:** Anthropometric data and laboratory tests for 51 women followed up after gestational diabetes mellitus (GDM), divided into groups of ^a^ normal glucose tolerance *N* = 12, ^b^ diabetes mellitus *N* = 16, ^c^ impaired fasting glucose (IFG) *N* = 13, and ^d^ impaired glucose tolerance (IGT) *N* = 10.

Variable	^a^ Normal *N* = 12	^b^ Diabetes *N* = 16	^c^ IFG *N* = 13	^d^ IGT *N* = 10
Weight before pregnancy (kg)	73.3 ± 14.4	79.1 ± 21.8	79.5 ± 20.9	73.9 ± 14.4
Weight at follow-up (kg)	74.4 ± 10	80.8 ± 18.5	83.4 ± 21.7	81.2 ± 17.9
Age (years)	45 ± 7	42 ± 6	41 ± 4	45 ± 6
Height (cm)	165 ± 5.3	166 ± 8.7	164 ± 6.6	164 ± 5.8
BMI (kg/m^2^)	26.9 ± 3.4	29.2 ± 5.8	31.1 ± 8.1	30.1 ± 5.6
Waist	91.9 ± 11.9	96.9 ± 14.4	99.7 ± 17.1	104.7 ± 17.7
Sagittal diameter (cm)	21.0 ± 2.9	22.4 ± 2.9	23.3 ± 4.7	24.3 ± 4.6
Systolic BP supine (mmHg)	116 ± 11	128 ± 15 *	124 ± 7	134 ± 20 *
Diastolic BP supine (mmHg)	73 ± 7	81 ± 11	79 ± 6	84 ± 13.2
Heart rate supine (beat/min)	66 ± 8	68 ± 11	64 ± 6	70 ± 8
P-Cholesterol (mmol/L)	5.2 ± 1.2	4.6 ± 0.99	5.0 ± 0.81	5.2 ± 0.89
P-LDL-cholesterol (mmol/L)	3.4 ± 1.1	2.8 ± 0.86	3.0 ± 0.82	3.3 ± 0.75
P-HDL-cholesterol (mmol/L)	1.4 ± 0.31	1.2 ± 0.35	1.4 ± 0.36	1.3 ± 0.20
P-Triglycerides (mmol/L)	1.1 ± 0.80	1.3 ± 0.63	1.17 ± 0.48	1.32 ± 0.60
APOB/APOA1	0.71 ± 0.24	0.67 ± 0.16	0.74 ± 0.21	0.80 ± 0.17
Breast feeding (%)	75% (*n* = 9)	56% (*n* = 9)	85% (*n* = 11)	80% (*n* = 8)
Pregnancies (*n*)	2.5 ± 0.5	2.2 ± 0.8	2.2 ± 0.8	3.5 ± 2.4
Birth weight **^¥^**	3053 ± 342	3462 ± 755	3162 ± 414	3381 ± 538
Heredity for diabetes **^†^**	67% (*n* = 8)	100% (*n* = 16) *	77% (*n* = 10)	80% (*n* = 8)

Data shown as mean ± SD or percentages. Comparisons (Students *t*-test or Chi-square test) between subjects with normal glucose tolerance and the other groups. * *p* <0.05. **^¥^** Own birth weight of the GDM subject. **^†^** First- and second-degree relatives combined. BP = blood pressure. BMI = body mass index. LDL = low density lipoprotein, HDL = high density lipoprotein. APOB/APOA1 = Apolipoprotein B/Apolipoprotein A1 ratio.

**Table 2 ijms-19-03724-t002:** Tests of glucose metabolism in 51 women at initial diagnosis of GDM and after 12 years. Follow-up divided into groups of ^a^ normal glucose tolerance *N* = 12, ^b^ diabetes mellitus *N* = 16, ^c^ impaired fasting glucose (IFG) *N* = 13, and ^d^ impaired glucose tolerance (IGT) *N* = 10. Glucose is reported as capillary blood glucose at GDM diagnosis and capillary plasma glucose at the clinical follow-up. Proinsulin was also transformed to its corresponding natural logarithm (ln) to accomplish a normal distribution.

Variable	^a^ Normal *N* = 12	^b^ Diabetes *N* = 16	^c^ IFG *N* = 13	^d^ IGT *N* = 10
At diagnosis of gestational diabetes mellitus (GDM) (capillary blood glucose)				
Fasting B-Glucose (mmol/L)	5.7 ± 0.8	5.6 ± 1.1	5.3 ± 0.9	5.3 ± 0.7
2 h 75 g OGTT before (mmol/L)	11.1 ± 0.9	11.5 ± 1.3	10.9 ± 0.7	10.6 ± 0.5
At clinical follow-up (capillary plasma glucose)				
Fasting P-Glucose (mmol/L)	5.6 ± 0.4	7.2 ± 1.5 **	6.5 ± 0.2	6.1 ± 0.7
2 h OGTT after (mmol/L)	6.4 ± 1.9	12.7 ± 3.5 ***	7.7 ± 1.0	10.0 ± 0.9 *
HbA1c, NGSP (%)	5.5 ± 0.3	6.3 ± 1 *	5.5 ± 0.2	5.6 ± 0.5
HbA1c, IFCC (mmol/mol)	37 ± 3	45 ± 12 *	37 ± 2	38 ± 5
C-peptide (nmol/L)	0.58 ± 0.29	0.54 ± 0.32	0.74 ± 0.32	0.73 ± 0.36
Proinsulin (pmol/L)	7.3 ± 8.1	10.9 ± 14.6	8.6 ± 6.7	9.7 ± 8.7
ln(Proinsulin) (pmol/L)	1.6 ± 0.8	1.9 ± 0.9	1.9 ± 0.8	1.9 ± 0.9
Ratio ln(Proinsulin)/C-peptide	2.9 ± 1.0	3.9 ± 1.7	2.6 ± 0.8	2.7 ± 1.0

Data shown as mean ± SD. Comparisons (ANOVA with Bonferroni post hoc test, *p* < 0.05) between subjects with normal glucose tolerance and the three other groups. *** *p* < 0.001, ** *p* < 0.01, * *p* < 0.05. OGTT = oral glucose tolerance test. HbA1c, NGSP = National Glycohemoglobin Standardization Program. HbA1c, IFCC = International Federation of Clinical Chemistry and Laboratory Medicine.

**Table 3 ijms-19-03724-t003:** Data shown for 51 subjects at follow-up 12 years after diagnosed gestational diabetes mellitus. Pearson correlation analyses (r) were performed between demographic/metabolic variables and ln(proinsulin) and C-peptide. Proinsulin was transformed to its corresponding natural logarithm (ln) to accomplish a normal distribution. Plasma glucose was measured using the capillary technique.

Variable	C-Peptide nmol/L		ln(Proinsulin) pmol/L	
	r	*p*	r	*p*
Age (yrs.)	0.02	n.s	0.13	n.s
Weight (kg)	0.48	<0.001 *******	0.50	<0.001 *******
BMI (kg/m^2^)	0.52	<0.001 *******	0.54	<0.001 *******
Waist circumference (cm)	0.60	<0.001 *******	0.60	0.001 *******
Sagittal diameter (cm)	0.51	0.001 *******	0.55	0.001 *******
Systolic BP mmHg)	0.40	0.003 ******	0.44	<0.001 *******
Diastolic BP (mmHg)	0.36	0.009 ******	0.35	0.013 *****
HbA1c (mmol/mol)	0.024	n.s	0.31	0.028 *****
Fasting P-glucose (mmol/L)	0.11	n.s	0.29	0.045 *****
2 h OGTT P-glucose (mmol/L)	0.14	n.s	0.37	<0.01 ******
P-Cholesterol (mmol/L)	0.19	n.s	0.23	n.s
P-HDL-Cholesterol (mmol/L)	−0.21	n.s	−0.40	0.004 *****
P-LDL-Cholesterol (mmol/L)	0.16	n.s	0.20	n.s
P-Triglycerides (mmol/L)	0.39	0.005 ******	0.48	<0.001 *******

******p* < 0.05**, ****
*p* < 0.01, *******
*p* < 0.001, n.s. = not significant

**Table 4 ijms-19-03724-t004:** Comparison of baseline data from 195 women diagnosed with gestational diabetes mellitus in the southeast region of Sweden during 1995–1999. In the present study, we invited women from the original cohort of 195 individuals living within 100 km of Linköping University Hospital for clinical examination; 51 accepted the invitation to participate in the study.

Parameter	Clinical Evaluation *N* = 51	Not Investigated *N* = 144	*p*-Value
Born in Sweden (%)	88	72	n.s.
Born Outside EU (%)	10	22	n.s.
Height (m)	1.65 ± 0.07	1.65 ± 0.07	n.s.
Weight (kg)	77 ± 19	74 ± 17	n.s.
BMI (kg/m^2^)	28 ± 6	28 ± 6	n.s.
OGTT 2 h Glucose (mmol/L)	9.9 ± 0.9	10.5 ± 1.7	0.032
Birthweight of the Child (g)	3664 ± 611	3624 ± 5696	n.s.

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
