# Peer review of "Most Women with Previous Gestational Diabetes Mellitus Have Impaired Glucose Metabolism after a Decade"

_ijms, 2018, doi:10.3390/ijms19123724_

Round 1

Reviewer 1 Report

Thank you for the opportunity to comment on this interesting paper. Although the cohort is rather small, I think it provides additional interesting information for the literature with its detailed description of the participants of the study population. Additionally, the discussion nicely deals with the limitations of the study and the generalizability of its results.

The readability of the manuscript could somewhat be improved, however. Please find my detailed comments below.

Abstract:

1. Page 1, line 16-17: a 2-h 75-g OGTT?

Introduction:

1. Page 1, line 32: I would perhaps consider again the use of the word “recent” when referring to a publication from already a decade ago.

Results:

1. The results section, more specifically its readability, could be improved by arranging the order of paragraphs somewhat. Currently one has to jump quite a lot back and forth while reading.

As an example, it would, in my opinion, be logical to move the paragraphs that define the different groups (i.e., paragraphs 2.4 and 2.6) closer to the beginning of the results section. It would be important to report the prevalence of these different glucose abnormalities before there are any references to the groups divided according to these abnormalities or to tables that include these groups (especially as the methods section is the last one, and therefore does not explain or define these groups already before the results section).

Another example is that when first reading paragraph 2.2 (page 3, lines 4-5) I was wondering if there was any difference between the four groups on heredity, i.e., the rate of relatives with DM. The answer at least partially comes in paragraph 2.5, that is, however, half a page further in the manuscript.

2. Page 2, lines 18-20: This sentence seems to imply, that none of the women had only one pregnancy, i.e., the index pregnancy? Or do these numbers include the index pregnancy?

3. Page 3, lines 20 and 21: compared to all other participants or women with normal glucose tolerance? These comparisons could probably be additionally clarified in other instanced in the manuscript, at least if the groups compared are different in different parts of the results.

4.Page 3, lines 31-32: On what data/finding is this sentence based?

5.Tables 1 and 2: The definitions for abbreviations used in the tables are mostly missing.

Results/Discussion:

1. As 2-h capillary blood glucose, ≥9.0mmol/l (≈plasma glucose ≥10.0mmol/l) is a rather high cut-off for GDM according to most current GDM criteria. I was wondering how many of the women did fulfill the criteria for T2D already during pregnancy?  Additionally, considering the rather high cut-off used for GDM, it would perhaps not have been a surprise, if a decade later there would have been even more T2D cases?

2. Page 5, lines 33-35: I find this comment a little vague, please clarify what results and how they might be explained?

Statistical analysis

1. Page 8, line 8 and Table 2: If I understand correctly ANOVA is used to compare differences between all the four groups. Were any post hoc procedures used or considered (Bonferroni, Tukey's HSD, etc.)? How were the comparisons between NGT and others done after ANOVA (Table 2)? And why did you choose to assess only the difference between NGT and others?

Author Response

Reviewer 1

Comments and Suggestions for Authors

Thank you for the opportunity to comment on this interesting paper. Although the cohort is rather small, I think it provides additional interesting information for the literature with its detailed description of the participants of the study population. Additionally, the discussion nicely deals with the limitations of the study and the generalizability of its results.

The readability of the manuscript could somewhat be improved, however. Please find my detailed comments below

Abstract:

1. Page 1, line 16-17: a 2-h 75-g OGTT?

Answer: This is correct. We have clarified that it was a standard “2-h 75-g OGTT”

Introduction:

1. Page 1, line 32: I would perhaps consider again the use of the word “recent” when referring to a publication from already a decade ago.

Answer: We agree and omit the word “recent” and replace it with “a meta-analysis in 2009 reported”

Results:

1.The results section, more specifically its readability, could be improved by arranging the order of paragraphs somewhat. Currently one has to jump quite a lot back and forth while reading.

Answer: We agree that we can make some changes in the paragraph order as the methodological section is the last one.

As an example, it would, in my opinion, be logical to move the paragraphs that define the different groups (i.e., paragraphs 2.4 and 2.6) closer to the beginning of the results section. It would be important to report the prevalence of these different glucose abnormalities before there are any references to the groups divided according to these abnormalities or to tables that include these groups (especially as the methods section is the last one, and therefore does not explain or define these groups already before the results section).

Answer: We took your advice and move the paragraph 2.4 and 2.6 closer to the beginning of the result section.

Another example is that when first reading paragraph 2.2 (page 3, lines 4-5) I was wondering if there was any difference between the four groups on heredity, i.e., the rate of relatives with DM. The answer at least partially comes in paragraph 2.5, that is, however, half a page further in the manuscript.

Answer: Thanks for this suggestion. We have now added data about heredity for all 4 groups in table 1 and if compared with Chi-squared test there was a difference between the subjects with normal glucose tolerance and the subjects with diabetes. This is added to the results section.

2. Page 2, lines 18-20: This sentence seems to imply, that none of the women had only one pregnancy, i.e., the index pregnancy? Or do these numbers include the index pregnancy?

Answer: Excuse us for our misspelling. These numbers include the index pregnancy. The number of pregnancies including the index pregnancy was one (n=4, 8%), two (n=25, 49%), three (n=18, 35%), four (n=3, 6%) and 10 pregnancies (n=1, 2%). This is corrected.

3. Page 3, lines 20 and 21: compared to all other participants or women with normal glucose tolerance? These comparisons could probably be additionally clarified in other instanced in the manuscript, at least if the groups compared are different in different parts of the results.

Answer: We have clarified this section. To have a relative with diabetes mellitus was more common in the group with diabetes compared with subjects with normal glucose tolerance.

4. Page 3, lines 31-32: On what data/finding is this sentence based?

Answer: The development of diabetes mellitus after GDM was not related to treatment they had during gestational diabetes period (diet or insulin).

5. Tables 1 and 2: The definitions for abbreviations used in the tables are mostly missing.

Answer: We have added definitions for abbreviations in the tables 1 and 2.

Results/Discussion:

As 2-h capillary blood glucose, ≥9.0mmol/l (≈plasma glucose ≥10.0mmol/l) is a rather high cut-off for GDM according to most current GDM criteria. I was wondering how many of the women did fulfill the criteria for T2D already during pregnancy?  Additionally, considering the rather high cut-off used for GDM, it would perhaps not have been a surprise, if a decade later there would have been even more T2D cases?

Answer: We agree of that. The new criteria for GDM, however, it is defined as a high risk for a large offspring. The original reason for defining and detecting GDM was to be able to identify those women at risk of diabetes mellitus later in life, and it was this hypothesis we tested. We fully agree that in future studies the lower cut of GDM has to be tested.

Page 5, lines 33-35: I find this comment a little vague, please clarify what results and how they might be explained?

Answer: We have clarified this sentence. “Despite women with known typ1 diabetes were excluded from the study the finding of further GAD positive women strengthens the data that around 5-6% in a Scandinavian GDM population are GAD positive postpartum”

Statistical analysis

1.Page 8, line 8 and Table 2: If I understand correctly ANOVA is used to compare differences between all the four groups. Were any post hoc procedures used or considered (Bonferroni, Tukey's HSD, etc.)? How were the comparisons between NGT and others done after ANOVA (Table 2)? And why did you choose to assess only the difference between NGT and others?

Answer: ANOVA was performed and Bonferroni post hoc test was used. We focused on the differences between the group with normal glucose tolerance and the other three groups as the main research question was to investigate all kind of glucose disturbances (ie, diabetes, IFG and IGT).

Reviewer 2 Report

1. Subchapter 2.4, line 22. It is: "... was Sweden (), Europe () and outside Europe ()" Change please this sentence. According to my knowledge, Sweden is in Europe. Therefore write name of country, not "Europe". Similar "outside Europe" see later.

2. Subchapter 2.7 Page 6, line 4. The sentence "... born outside Europe..." is unclear. There are different risk of development diabetes mellitus in dependence on ethinicty. It is different for people from South Africa, Nord America, South East Asia etc. Write please "where these women from". Maybe this has influence on results.

3. Minor suggestions:

a) Why "2-h 75-g OGGT" is "a new"? This information is for example in Abstract as well as in later parts of manuscript.

b) Write please 0.5% not) 0.5 % (without space) - few times in text.

c) On the the other hand, there are sentences where spaces are necessary. For example: "... positive postpartum [27]." It should be postpartum [27]

d) It is unclear " as mean + 1 SD" It should be without "1"

e) Page 5 line 46. In Finish population is observed high prevalence of type 1 diabetes.

Author Response

Reviewer 2

Comments and Suggestions for Authors

1. Subchapter 2.4, line 22. It is: "... was Sweden (), Europe () and outside Europe ()" Change please this sentence. According to my knowledge, Sweden is in Europe. Therefore write name of country, not "Europe". Similar "outside Europe" see later.

Answer: We have clarified this sentence and added more information of origin. “The country of origin was Sweden (88%, n=45), Europe outside Sweden (2%, n=1) and outside Europe (10 %, Middle East n=4 and China n=1)”.

2. Subchapter 2.7 Page 6, line 4. The sentence "... born outside Europe..." is unclear. There are different risk of development diabetes mellitus in dependence on ethinicty. It is different for people from South Africa, Nord America, South East Asia etc. Write please "where these women from". Maybe this has influence on results.

Answer: Se answer question 1.

3. Minor suggestions:

a) Why "2-h 75-g OGGT" is "a new"? This information is for example in Abstract as well as in later parts of manuscript.

Answer: We have clarified that it is a second 2 h 75 g OGGT to discriminate from the first OGTT diagnostic for the GDM diagnosis.

b) Write please 0.5% not) 0.5 % (without space) - few times in text.

Answer: Thanks for these corrections. We have changed this in the manuscript.

c) On the the other hand, there are sentences where spaces are necessary. For example: "... positive postpartum [27]." It should be postpartum [27]

Answer: Thanks for these corrections. We have changed this in the manuscript.

d) It is unclear ". as mean + 1 SD" It should be without "1"

Answer: We have corrected this.

e) Page 5 line 46. In Finish population is observed high prevalence of type 1 diabetes.

Answer: That is correct that it is common with type 1 diabetes in Finland. One could therefore suggest that lean Finnish women diagnosed with diabetes mellitus after GDM much likely have type 1 diabetes.

Reviewer 3 Report

Please see my comments below:

1) Please revise the introduction. Please introduce the topic and explain the reasons why the topic is important for the research study. This is not well explained in the introduction. The current version of introduction is too brief and does not address these isses.

2) Introduction: Why are the limitations of study discussed here? It doesn't make sense.

3) Section 4: Please revise these paragraphs by removing the Step 1 and Step 2. They needs to look scientific.

4) Section 4.2: Please provide the categorisation of blood pressure measurement after the measurement of blood pressure. It shouldn't be in the same sentence as the obesity cut-off.

5) Section 4.4: What was the p value used for the Bonferroni post hoc tests?

6) Discussion section: the authors should compare the results with previous studies and highlight the major findings of the study. Although the authors have done these, these are not sufficient enough because the authors did not criticially appraise the evidence enough. What is seen from the discussion is that the authors only explained the results and discussed the results descriptively. Therefore, the authors should try to revise this section.

Author Response

Reviewer 3

Comments and Suggestions for Authors

Please see my comments below:

1) Please revise the introduction. Please introduce the topic and explain the reasons why the topic is important for the research study. This is not well explained in the introduction. The current version of introduction is too brief and does not address these isses.

Answer: We have expanded the introduction at page 1-2 and added 3 more references (8-10).

8. Lawrence, J. M.; Contreras, R.; Chen, W.; Sacks, D. A., Trends in the prevalence of preexisting diabetes and gestational diabetes mellitus among a racially/ethnically diverse population of pregnant women, 1999-2005. Diabetes Care 2008, 31, (5), 899-904.

9. Ostlund, I.; Hanson, U.; Bjorklund, A.; Hjertberg, R.; Eva, N.; Nordlander, E.; Swahn, M. L.; Wager, J., Maternal and fetal outcomes if gestational impaired glucose tolerance is not treated. Diabetes Care 2003, 26, (7), 2107-11.

10. Kessous, R.; Shoham-Vardi, I.; Pariente, G.; Sherf, M.; Sheiner, E., An association between gestational diabetes mellitus and long-term maternal cardiovascular morbidity. Heart 2013, 99, (15), 1118-21.

2) Introduction: Why are the limitations of study discussed here? It doesn't make sense.

Answer: This is not a limitation of the current study. We have included this section in the introduction as it is important to understand that with a questionnaire we cannot find out all glucose abnormalities as they are not diagnosed yet. We have rephrased this section to a more general approach at page 1-2 lines 43-46:

3) Section 4: Please revise these paragraphs by removing the Step 1 and Step 2. They needs to look scientific.

Answer: We did not fully understand the question, but agree that we can remove the initial words, Step 1 and Step 2.

4) Section 4.2: Please provide the categorisation of blood pressure measurement after the measurement of blood pressure. It shouldn't be in the same sentence as the obesity cut-off.

Answer: thanks for this correction. We have changed this sentence as suggested.

5) Section 4.4: What was the p value used for the Bonferroni post hoc tests?

Answer: The adjusted p value after Bonferroni post hoc tests was p<0.05.< span="">

6) Discussion section: the authors should compare the results with previous studies and highlight the major findings of the study. Although the authors have done these, these are not sufficient enough because the authors did not criticially appraise the evidence enough. What is seen from the discussion is that the authors only explained the results and discussed the results descriptively. Therefore the authors should try to revise this section.

Answer: We agree that our results could be even more highlighted, and we have added a section about the importance of IGT and IFG to the discussion section.

“There is no consensus regarding the use of fasting plasma glucose, OGTT, or HbA1c when following up on GDM. Glucose abnormalities defined by both IFG and IGT are risk predictors for diabetes, even if IGT defines a much larger target population for prevention [17]. It should be noted that in our study, HbA1c could not distinguish between normal subjects and subjects with IGT and/or IFG. However, in the group with diabetes, HbA1c was elevated; it should therefore be considered for follow-up and is today generally recommended for diagnosis of diabetes [18]. Overall, the high frequency of glucose abnormalities found in this and previous studies indicate that women with GDM should be offered lifelong regular glucose measurements.”

17. Qiao, Q.; Lindstrom, J.; Valle, T. T.; Tuomilehto, J., Progression to clinically diagnosed and treated diabetes from impaired glucose tolerance and impaired fasting glycaemia. Diabet Med 2003, 20, (12), 1027-33.

18. American Diabetes, A., 2. Classification and Diagnosis of Diabetes: Standards of Medical Care in Diabetes-2018. Diabetes Care 2018, 41, (Suppl 1), S13-S27.

Round 2

Reviewer 1 Report

Although most of my concerns have been dealt with, I continue to hope that the readability/fluency/language of the manuscript would be improved.

Author Response

Reviewer 1

Comments and Suggestions for Authors

Although most of my concerns have been dealt with, I continue to hope that the readability/fluency/language of the manuscript would be improved.

Answer: The manuscript has been now been checked for language and layout by MDPI English Editing service. We hope the additional changes we have made to the manuscript, in accordance with the comments by the other referees, and the language and layout checks make the paper suitable for publication.